# Automatic Fault Detection and Isolation Method for Roller Bearing Using Hybrid-GA and Sequential Fuzzy Inference ^†^

**DOI:** 10.3390/s19163553

**Published:** 2019-08-15

**Authors:** Yusuke Kobayashi, Liuyang Song, Masaru Tomita, Peng Chen

**Affiliations:** 1Railway Technical Research Institute, Materials Technology Division, Applied Superconductivity Laboratory, Tokyo 185-8540, Japan; 2College of Mechanical and Electrical Engineering, Beijing University of Chemical Technology, Beijing 100029, China; 3Graduate School of Environmental Science and Technology, Mie University, 1577 Kurimamachiya-cho, Tsu, Mie 514-8507, Japan

**Keywords:** condition diagnosis, bearing faults, hybrid-GA, noise cancelling, fuzzy inference

## Abstract

Though accelerometers for condition diagnosis of a bearing is preferably placed at the nearest position of the bearing as possible, in some plant equipment, the accelerometer is difficult to set near the diagnosed bearing, and in many cases, sensors have to be placed at a location far from the diagnosed bearing to measure signals for diagnosing bearing faults. Since, in these cases, the measured signals contain stronger noise than the signal measured near the diagnosed bearing, bearing faults are more difficultly to be detected. In order to overcome the above difficulty, this paper proposes a new fault auto-detection method by which the signals measured by an accelerometer located at a far point from the diagnosed bearing can be used to simply and accurately detect the bearing faults automatically. Firstly, the hybrid GA (the combination of genetic algorithm and tabu search) is used to automatically search and determine the optimum cutoff frequency of the high-pass filter to extract the fault signal of the abnormal bearing. Secondly, the bearing faults are precisely diagnosed by possibility theory and fuzzy inference. Finally, in order to demonstrate the effectiveness of these proposed methods, these methods were applied to bearing diagnostics using vibration signals measured at the far point of the diagnostic bearing, and the efficiency of these methods was verified by the results of automatic bearing fault diagnosis.

## 1. Introduction

The bearing is the most important component to support the rotating mechanical shafting [1]. Bearing failures can be classified into initial failures (point injuries caused by local spalling), medium-term failures (large-scale peeling of inner and outer rings), and final failures (fracture of rolling elements and cages) [2,3]. The final failure of the bearing often leads to the failure of the rotating shaft and causes a major accident. Therefore, it is very important to effectively detect the bearing failure as soon as possible before the final bearing failure.

After the bearing is damaged, a pulsating vibration occurs due to the rolling element passing through the damage [4]. This kind of pulsating vibration is the characteristic resonance of the inner or outer ring of the bearing. The frequency of this characteristic resonance is also called the “ringing frequency”. The ringing frequency is generally in the high frequency band, higher than several thousand Hz, depending on the type and Size of bearing [5]. To detect incipient bearing abnormality by vibration diagnosis, it is important to sense this impulsive vibration wave in the high frequency band and cancel noise by high-pass-filter (HPF) [6]. Several filter parameters optimization methods have been summarized to extract fault feature from signals of the bearing with incipient fault [7,8]. Wang et al. proposed a complex Morlet wavelet filter which parameters are optimized by simplex-simulated annealing [9]. Jia et al. constructed a high-pass filter based on banded matrices to diagnosis incipient bearing fault efficiency [10]. A motor fault diagnosis method based on a simple nonlinear observer was proposed, and the effectiveness of the method is confirmed by the experiment with a permanent magnet direct current (DC) motor [11]. It was evidenced that in the fault diagnosis methods based on principal components analysis(PCA), the number of the principal components to be retained and the data matrix is important which will affect the performance of the algorithm [12]. Deep Boltzmann machines, deep belief networks and stacked auto-encoders were applied to diagnose the fault of the rolling bearing [13]. The authors pointed out that feature extraction of the vibration signal is a necessary step for the deep neural networks based (DNN -based) classifiers which can incline the reliability and effectiveness of the fault diagnosis method. The above fault diagnosis methods have certain effects in an off-line diagnosis system. However, when diagnosing bearing faults, if there is a strong noise in the diagnostic signals, such as an early bearing abnormality, or the sensor cannot be placed near the diagnostic object bearing to measure signals, etc., how to extract weaker fault signal from strong noise signal, and how to deal with ambiguous diagnostic information in real plant diagnosis still have many problems to be solved. Furthermore, because when the bearing has an early abnormality (spot injury), it takes a long time (usually several days to several months) to progress to the mid-term anomaly stage (inner and outer ring flaking), so online real-time diagnosis is not required when diagnosing a bearing fault. As long as the fault can be detected after measuring a vibration signal, then the fault type can be identified. Therefore, this paper proposes a method for data training and a fuzzy diagnostic algorithm that can be embedded in the offline bearing diagnostic system.

When identifying the roller bearing anomaly of the field device by the vibration signal, there are the following two problems:

(1) It is needed to decide a cutoff frequency of HPF to extract a fault signal. However, the ideal cutoff frequency is different because the characteristic frequency of inner or outer rings of bearings is different by the type or size of the bearings. Therefore, there is a need to establish a method for automatically determining the optimum cutoff frequency that can be used in a bearing diagnostic system, and the method must have versatility independent on the tested device.

(2) To catch abnormal vibration signals of bearings sensitively, it is necessary to set accelerometers as near to the diagnosed bearings as possible. However, in some field equipment, it is difficult to set accelerometers near the diagnose bearings, the vibration signals can only be measured at a position far from the bearing. Since, in these cases, the measured signals contain stronger noise than the signal measured near the diagnosed bearing, it is more difficult to diagnose the bearing faults.

In this research, to solve the problems described above, novel methods by which bearing faults can be automatically, simply and precisely diagnosed using the signals measured by the acceleration sensors at distant points from the diagnosed bearings was proposed as follows: 

(1) In order to extract the characteristic signal of a fault bearing from the signal measured far from the bearing as sensitively as possible, a method by which the optimal cutoff frequency of HPF is automatically searched and decided by hybrid GA (the combination of genetic algorithm and tabu search, GA+TS) is proposed.

(2) Since the noise in the measured signals for diagnosing the bearing cannot be completely cancelled, and due to the influence of the degree of bearing damage, there is often an ambiguity when diagnosing the bearing fault. In order to solve this problem, a sequential fuzzy diagnosis method is proposed by which the bearing faults can be automatically and sequentially diagnosed. 

(3) In order to effectively carry out the sequential fuzzy diagnosis of a bearing, new symptom parameters specific to diagnose bearing are defined, and the method of defining the membership function of these symptom parameters by using possibility function is also proposed.

(4) Finally, these methods were applied to bearing diagnosis using the vibration signals measured at distant points from the diagnosed rolling bearings, and the efficiency of these methods have been verified by the results of the automatic bearing faults diagnosis. 

## 2. The Principle of Bearing Diagnosis

### 2.1. Entire Flowchart

The genetic algorithm (GA) has a good global search ability, and the tabu search has a good local search ability [14,15]. A number of fault diagnosis methods based on GA [16,17,18] and tabu search [19,20] have been proposed. In the proposed method, in order to effectively extract a vibration signal from an abnormal bearing, the cutoff frequency of a high-pass-filter is automatically optimized by hybrid GA (genetic algorithm combining with tabu search algorithm). Moreover, bearing faults are precisely diagnosed by fuzzy inference, in which the fault detection step and the fault isolation step are included. The proposed methods were applied to bearing diagnosis using the vibration signals measured at distant points from the diagnosed bearing, and the effectiveness have been verified by the results of the automatic bearing fault diagnosis. A block diagram is shown in Figure 1 to explain the entire process of the proposed methods. 

(1) Fault detection:

In the field of equipment condition diagnosis, fault detection also known as “simple diagnosis” and refers to the use of methods to determine whether an equipment is normal or abnormal [21,22].

In the fault detection step of a bearing, the kurtosis shown in (1) is used to determine whether the bearing is abnormal. Furthermore, the kurtosis is also used as an indicator to search the optimum cutoff frequency of the HPF for extracting fault signal.

(2) Fault isolation:

The precision diagnosis is used to identify the type of abnormality after the equipment state is monitored as abnormal by the fault detection. Since the noise in the measured signals for diagnosing the bearing cannot be completely cancelled, and due to the influence of the degree of bearing damage, there is often an ambiguity when diagnosing the bearing fault.

In order to precisely diagnose bearing faults, the symptom parameters specific to diagnose bearing are defined, and the sequential fuzzy diagnosis is carried out by using these symptom parameters. The membership functions of these symptom parameters are obtained by using the possibility function.

### 2.2. Fault Detection

In the fault detection of the bearing, the kurtosis shown in (1) is used to determine whether the bearing is abnormal. Here, *x_i_* (*i* = 1~N) are time series discrete acceleration data. The kurtosis ratio (RK) in the diagnosis stage and in normal status as shown in (4) is used to judge the state of the diagnosed bearing. In order to cancel the noise from the measured signal for the diagnosis, a high-pass filter (HPF) is used in the range between 0 and 40,000 Hz, and its optimum cutoff frequency can be decided by using hybrid GA (GA and TS) to maximize the RK. That is to say, the RK is used as fitness when exploring the optimum cutoff frequency with GA both in the learning phase and diagnosis phase.
(1)$$Kurtosis=\frac{\sum_{i=1}^{N}{\left(x_{i}-\bar{x}\right)}^{4}}{N\sigma^{4}}$$
(2)$$\bar{x}=\frac{\sum_{i=1}^{N}x_{i}}{N}$$
(3)$$\sigma =\sqrt{\frac{\sum_{i=1}^{N}{\left(x_{i}-\bar{x}\right)}^{2}}{N-1}}$$
(4)$$RK=\frac{Kurtosis_{Unknown}}{Kurtosis_{Normal}}$$

Here, *Kurtosis_Normal_* is the kurtosis in the normal state, and *Kurtosis_Unknown_* is the kurtosis in the diagnosis step. In the process of genetic algorithm optimization, the cutoff frequency is converted to a binary number of length 8 as a gene, the gene number is 10, the mutation rate is 0.7, and the crossover rate is 0.1. A tabu search (TS) is also introduced because the use of GA alone may lead to a local optimal solution, and the method of combining GA and TS is called “Hybrid-GA” in this paper. That is, the frequency near the optimum cutoff frequency obtained by the GA is searched under the taboo condition that the frequency once selected is never selected again. In this case, in the ± 10,000 Hz range of the optimum cutoff frequency searched by the GA, the search for a better cutoff frequency (Figure 1) is carried out without repeating in step of 1,000 Hz (Figure 1), and if the RK at the cutoff frequency exceeds the RK at the old optimum cutoff frequency, the new optimum cutoff frequency is regained. Whether the bearing is normal or abnormal can be determined by whether the RK obtained by the above method is greater than the threshold (how to determine the threshold will be described in detail in Section 3.2).

### 2.3. Fault Isolation

#### 2.3.1. Flow of the Fault Isolation

Figure 2 shows the flowchart of the bearing fault isolation, and the detail process will be explained in this section. In the learning phase: (1) Measuring vibration signals in various states (normal, outer race flaw, inner race flaw and rolling element flaw), and the bearing flaws are caused by using wire cutting. Learning data and the verification data are measured separately; (2) calculating the spectra of the raw signals (Fast Fourier Transform, FFT); (3) using hybrid GA for searching the optimum cutoff frequency to cancel noise; (4) calculating the envelope waveform of the signals after noise cancelling; (5) calculating the spectra of the envelope waveforms; (6) calculating the symptom parameters for bearing diagnosis (symptom parameters for bearing fault diagnosis on outer race, SPO and symptom parameters for bearing fault diagnosis on inner race, SPI); (7) obtaining membership functions that are used to identify the type of bearing fault by the possible functions of SPO and SPI. 

In the diagnostic phase: (1) Measuring vibration signal for diagnosis; (2) performing FFT processing on the original signal; (3) cancelling noise using the optimum cutoff frequency; (4) calculating the envelope of the signal after cancelling noise; (5) calculating the spectrum of the envelope waveform; (6) calculating the dedicated symptom parameters (SPO and SPI) for bearing diagnosis; (7) calculating the possibility functions of SPO and SPI; (8) the fault type of the bearing is identified by sequentially matching the possible functions of SPO and SPI with the membership functions of the bearing fault types.

#### 2.3.2. Dedicated Symptom Parameters for Bearing Diagnosis

In order to precisely diagnose bearing faults, the spectrum components of path frequency of shocking waveforms caused by bearing flaws are determined by the envelope spectrum of the vibration waveform to identify abnormality types. The path frequencies of bearing flaws are calculated by following formula [23].
(5)$$f_{O}=\frac{zf_{r}}{2}(1-\frac{d}{D}\cos\alpha )\;(\mathrm{Outer}\;\mathrm{race}\;\mathrm{flaw})$$
(6)$$f_{I}=\frac{zf_{r}}{2}(1+\frac{d}{D}\cos\alpha )\;(\mathrm{Inner}\;\mathrm{race}\;\mathrm{flaw})\;$$
(7)$$f_{R}=\frac{Df_{r}}{2}(1-\frac{d^{2}}{D^{2}}\cos\alpha )\;(\mathrm{Rolling}\;\mathrm{element}\;\mathrm{flaw})$$
Here, *f_r_* is the rotating frequency; *z* is the roller number, *d* is the diameter of roller; *D* is the pitch diameter; α is the contact angle.

The signal measured by an acceleration sensor set at a distance from the diagnosed bearing is filtered using the optimal high-pass filter searched by GA, and the waveform after envelope processing is used for fault isolation. Using the spectrum of the envelope waveform, a new dedicated symptom parameter for bearing diagnosis reflecting the path frequency component of the bearing flaw is proposed as the following equation, and an automatic fault isolation method (automatic identification method of fault type) by sequential fuzzy reasoning is proposed.
(8)$$SPx=\frac{N\sum_{i=1}^{M}S_{fxi}}{\sum_{i=1}^{N}S_{j}}$$
Here, *S_j_* is the value of spectrum at the *j* (Hz) frequency, and *f_xi_* is the ith harmonic frequency of the pass frequency in x state. *i* = 1, 2, …, *M* (*M* > 2) and x = O, I, R (O: Outer race flaw, I: Inner race flaw, R: Rolling element flaw). $\sum_{i=1}^{N}S_{j}$ represents the integral value of the entire spectral interval, and $\sum_{i=1}^{M}S_{fxi}$ represents the sum of the harmonic frequency components of the pass frequency. The type of bearing fault is distinguished by using the DSP and sequential fuzzy inference. 

#### 2.3.3. Membership Function for Fuzzy Inference

For fuzzy inference, the membership functions of *SPx* is necessary to identify the bearing states. The membership function can be obtained from the probability density functions of the *SPx* using possibility theory. For example, when the probability density function of *SPx* conforms to the normal distribution, it can be changed to possibility functions P(*SPx*) by the following formulae [24,25].
(9)$$P(SPx)=\sum_{k=1}^{N}\min\{\lambda_{i},\lambda_{k}\}$$
here, $\;\lambda_{i}$ can be calculated as follows;
(10)$$\lambda_{i}=\int_{SPxi-1}^{SPxi}\frac{1}{\sigma SPx\sqrt{2\pi }}\exp\{-\frac{{\left(x-\bar{SPx}\right)}^{2}}{2\sigma SPx^{2}}\}dx$$
Here, $\sigma_{SPx}$ means the standard deviation, $\bar{SPx}$ means the mean value of SPx, and $SPx=\bar{SPx}-3\sigma_{spx}\sim \bar{SPx}+3\sigma_{spx}$.

An example of a probability density function and a possible function is shown in Figure 3. The membership function obtained by this method is objective because the membership function is obtained based on the mean and standard deviation of the actual field data.

As shown in Figure 4, the fault type of a bearing can be precisely diagnosed by sequential fuzzy inference. The membership functions for identifying fault types can be obtained by the possibility functions. The membership functions $\mu_{O}$ and $\mu_{\bar{O}}$ for identifying that the state is outer race flaw or not are defined as follows.
(11)$$\mu_{\bar{O}}(SP_{O})=\{\begin{array}{c} P^{\left(R\right)}(SP_{O})\cup P^{\left(I\right)}(SP_{O})\;\;\;\;\;\;\;\;\;SP_{O}\geq \max(\bar{SP_{O}^{\left(I\right)}},\;\bar{SP_{O}^{\left(R\right)}})\\ 1.0\;\;\;\;\;\;\;\;\;\;\;\;\;\;\;\;\;\;\;\;\;\;\;\;\;\;\;\;\;\;\;\;\;\;\;\;SP_{O}<\max(\bar{SP_{O}^{\left(I\right)}},\;\bar{SP_{O}^{\left(R\right)}}) \end{array}$$
(12)$$\mu_{O}(SP_{O})=1-\mu_{\bar{O}}(SP_{O})$$
Here, the meanings of possibility function $P^{\left(R\right)}(SP_{O}),\;\;P^{\left(I\right)}(SP_{O}),\;\;\bar{SP_{O}^{\left(I\right)}}\;\;and\;\;\bar{SP_{O}^{\left(R\right)}}$ to identify outer race flaw are as shown in Figure 5. 

Figure 5 also shows the membership functions obtained with signals measured by two sensors (CH1 and CH2), and the details will be discussed in the next section. 

The membership functions $\mu_{I}$ and $\mu_{\bar{I}}$ for identifying that the state is outer race flaw or not are defined as follows.
(13)$$\mu_{\bar{I}}(SP_{I})=\{\begin{array}{c} P^{\left(R\right)}(SP_{I})\cup P^{\left(O\right)}(SP_{I})\;\;\;\;\;\;\;\;\;SP_{I}\geq \max(\bar{SP_{I}^{\left(O\right)}},\;\bar{SP_{I}^{\left(R\right)}})\\ 1.0\;\;\;\;\;\;\;\;\;\;\;\;\;\;\;\;\;\;\;\;\;\;\;\;\;\;\;\;\;\;\;\;\;\;\;\;SP_{I}<\max(\bar{SP_{I}^{\left(O\right)}},\;\bar{SP_{I}^{\left(R\right)}}) \end{array}$$
(14)$$\mu_{I}(SP_{I})=1-\mu_{\bar{I}}(SP_{I})$$
Here, the meanings of possibility function $P^{\left(R\right)}(SP_{I}),\;\;P^{\left(I\right)}(SP_{I}),\;\;\bar{SP_{I}^{\left(I\right)}}\;\;and\;\;\bar{SP_{I}^{\left(R\right)}}$ to identify inner race flaw are shown in Figure 6. Figure 6 also shows the membership functions obtained with signals measured by two sensors (CH1 and CH2), and the details will be discussed in the next section.

#### 2.3.4. Sequential Fuzzy Diagnosis

The state set is $C_{x}$, here, $C_{x}=O$: Outer race flaw, $C_{x}=I$: Inner race flaw and $C_{x}=R$: Rolling element flaw. The state of the bearing to be determined is denoted as $C_{y}$, and as shown in Figure 4, the rule of sequential diagnosis inference for identifying the state $C_{y}$ is shown as follows.

Rule 1:Prerequisite 1: If $SPO>SPO^{*}$ then $C_{y}$ is $C_{O}$;Prerequisite 2: If $SPO<SPO^{*}$ then $C_{y}$ is $\bar{C_{O}}$;Input: SPO=SPO′Conclusion: $C_{y}$ is $C_{O}$ or $\bar{C_{O}}$;(If $C_{y}$ is $C_{O}$, then end of the inference.)
Rule 2:Prerequisite 1: If $SPI>SPI^{*}$ then $C_{y}$ is $C_{I}$;Prerequisite 2: If $SPI<SPI^{*}$ then $C_{y}$ is $C_{R}$;Input: SPI=SPI′Conclusion: $C_{y}$ is $C_{I}$ or $\bar{C_{R}}$;

Here, $SPO^{*}$ and $SPI^{*}$ are the threshold values for state judgement as shown in Figure 5 and Figure 6. SPO′ and SPI′ are the values calculated with the signals measured for bearing state diagnosis.

The membership function of the SPx′ obtained in the diagnosis stage is expressed as $\mu_{y}(SP_{x}\prime )$, and the membership degrees (possibilities) of $C_{k}$ and ${\bar{C}}_{k}$ obtained by $\mu_{y}(SP_{x}\prime )$ are respectively expressed as $w_{k}$, $w_{\bar{k}}$; The possibilities are obtained using the formulae (15) and (16): (15)$$w_{k}=\vee \{\mu_{C_{k}}(PS_{x}\prime )\wedge \mu_{y}(PS_{x}\prime )\}$$
(16)$$w_{\bar{k}}^{}=\vee \{\mu_{\bar{C_{k}}}^{}(PS_{x}\prime )\wedge \mu_{y}^{}(PS_{x}^{}\prime )\}$$

According to the flowchart in Figure 4, examples of bearing fault diagnosis results based on the method proposed in this paper are shown below to illustrate the effectiveness of the proposed method.

## 3. Verification

### 3.1. Rotating Machine for Verification Experiments 

Figure 7 shows a rotating machine for experiments, the rotating shaft of which has a maximum rotational speed up to 2000 rpm. Bearings can be replaced with normal and abnormal bearings, and abnormal bearing are outer race flaw, inner race flaw and rolling element flaw, as shown in Figure 8. Two accelerometers (Fuji Ceramics, SA12SC, sensitivity: 10 mV/g) were installed for measuring vibration signal, and CH1 was close to the bearing, and CH2, CH3 were 510 mm, 570 mm from the bearing. Acceleration signals at 1800 rpm were acquired at CH1, CH2 and CH3 at 100 kHz sampling frequency and 10s sampling time.

### 3.2. Learning Phase

To decide the RK threshold value for the fault detection of bearing, the experiment of the learning phase will be done in Section 3.2.1 and Section 3.2.2.

#### 3.2.1. Learning Step of Fault Detection

Table 1 shows the optimum cutoff frequency and RK obtained by the data sets of normal, outer race flaw, inner race flaw, and rolling element flaw of CH3 as an example, because signals of CH3 are collected at the point farthest from the fault bearing, which will increase the difficulty of diagnosis. From Table 1, the threshold of RK was determined to be 1.2, in order to determine whether the bearing is normal.

The optimum cutoff frequency and the change trend of RK in the case of the signals of the outer race flaw measured at CH1 are shown in Table 2. These data show an example where even though the result of the GA searching may be local extremum value, using hybrid GA to search again can jump over the local extremum value to find the maximum value. Figure 9 shows the relationship between the cutoff frequency and RK, and also shows a well example of the usefulness of hybrid GA (GA and TS).

#### 3.2.2. Learning Step of Fault Isolation

The process and results of learning and fault isolation according to Figure 4, Equations (8) to (16), and fuzzy inference (rule1, rule 2) are illustrated below by taking the data measured at CH3 as an example. Table 3 shows the DSPs shown in (8) for bearing diagnosis using the acceleration data measured at CH3 in the states of outer race flaw, inner race flaw and rolling element flaw. The measured data is equally divided into eight sections, and eight values of the DSPs are obtained for each fault type.

$\bar{SPx}$ and $\sigma_{SPx}$ of *SPx* are calculated with the signals measured in each state in the learning step as shown in Table 4. According to formulas (9), (11) and (12), the membership functions $\mu_{O}\left(SP_{O}\right)$ and $\mu_{\bar{O}}\left(SP_{O}\right)$ for identifying the outer race flaw are shown in Figure 10. In addition, according to formulas (9), (11) and (12), the membership functions $\mu_{I}\left(SP_{I}\right)$ and $\mu_{\bar{I}}\left(SP_{I}\right)$ are for identifying the outer race flaw are shown in Figure 11.

Figure 12 shows the results of filtering and envelope processing for the outer race flaw data measured at CH3 at 1800rpm. It can be seen from Figure 12d–f, that the noise in the low frequency band is removed in Figure 12f, and the spectra corresponding to the passing frequency (128.2Hz) and its harmonic frequencies caused by outer race flaw of the bearing are extracted. It also can be seen from Figure 12a–c, after a high-pass filter with the optimized cutoff frequency, the base noise is greatly eliminated, and the usefulness of the envelope process after the HPF using the optimum cutoff frequency can be confirmed.

### 3.3. Diagnosis Phase

The data that was not used in the learning step are used to verify the bearing diagnosis process discussed in Section 3.3.1 and Section 3.3.2. 

#### 3.3.1. Diagnosis Step of Fault Detection

The results of fault detection in the verification step are shown in Table 5. From Table 5, it can be seen that the RK in the normal bearing is smaller than the threshold value 1.3, and the RKs in the abnormal bearings are larger. By these results, the effectiveness of the fault detection method proposed in this paper for roller bearing is verified.

#### 3.3.2. Diagnosis Step of Fault Isolation

For fault isolation of the bearing in the verification step, the DSPs are calculated as shown in Table 6. From Table 6, $\bar{SPx}$ and $\sigma_{SPx}$ of *SPx* are calculated with the signals measured in each state as shown in Table 7.

According to formulas (11) to (14), the possibility function of the *SPx* obtained in the diagnosis step is written as $\mu_{y}$(SPx’), as shown in Figure 10 and Figure 11. The possibility functions of $\mu_{y}$(SPx’) shown in Figure 10 and Figure 11 are calculated using $\bar{SPx}$ and $\sigma_{SPx}$ shown in Table 6 for the verification. 

Table 8 and Table 9 show the fuzzy reference results based on the fuzzy inference rule 1 and 2, formula (15) and (16), Figure 10 and Figure 11. It can be seen from these results that the abnormal states have all been correctly diagnosed, and the validity of the bearing diagnosis method proposed in this paper was verified. 

In order to analyze the performance of the proposed method, the recognition accuracy of the proposed method is compared with the algorithms commonly used for fault diagnosis, such as artificial neural network (ANN) [26], support vector machine (SVM) [27], k-nearest neighbors (KNN) [28] and convolutional neural network (CNN) [29]. The comparison experiments are carried out by the MATLAB Statistics and Machine Learning Toolbox 11.1, which included implementations of various machine learning classifiers. The classifiers are trained by the measured signals and the verification results are compared in Figure 13.

## 4. Conclusions

This paper proposed a new fault auto-detection method for a roller bearing, by which the optimum cutoff frequency for removing noise from the vibration signal measured for the bearing diagnosis can be automatically searched and the state of the bearing can be automatically diagnosed. With this method, even if the accelerometer cannot be placed near the diagnosed bearing and is located away from the bearing to be diagnosed, it is also possible to diagnose the bearing fault more accurately.

In order to detect the bearing fault signals, a method of searching optimal cutoff frequency of high-pass filter by hybrid GA combining GA and TS is proposed, and RK is used as the fitness of GA, and the fault detection of the bearing is also carried out. In order to precisely determine the type of the bearing fault, the symptom parameters specific for bearing diagnosis are proposed, and the membership functions of the symptom parameters are established according to possibility theory for fuzzy diagnosis. After the bearing is judged as abnormality by the fault detection, the membership functions of the symptom parameters dedicated to bearing diagnosis and the fuzzy inference are used to sequentially and precisely identify the type of the bearing fault, and it has been confirmed that the feature vibration signal caused by bearing faults can be detected with an accelerometer set at a distant point from the diagnosed bearing by using the proposed algorithms.

From now on, we will do the same research on sliding bearings, aiming to construct useful high-precision sliding bearing diagnostic methods.

## Figures and Tables

**Figure 1 sensors-19-03553-f001:**
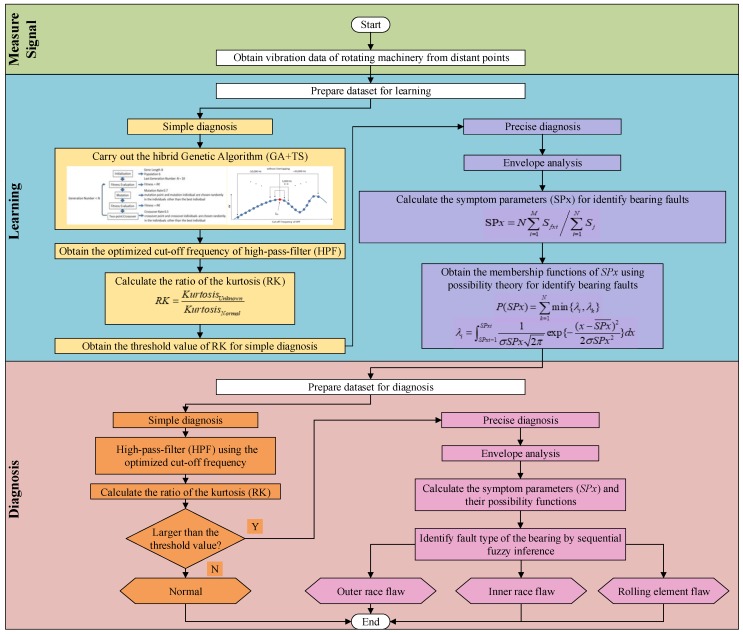
Flowchart of the diagnosis process proposed in this study.

**Figure 2 sensors-19-03553-f002:**
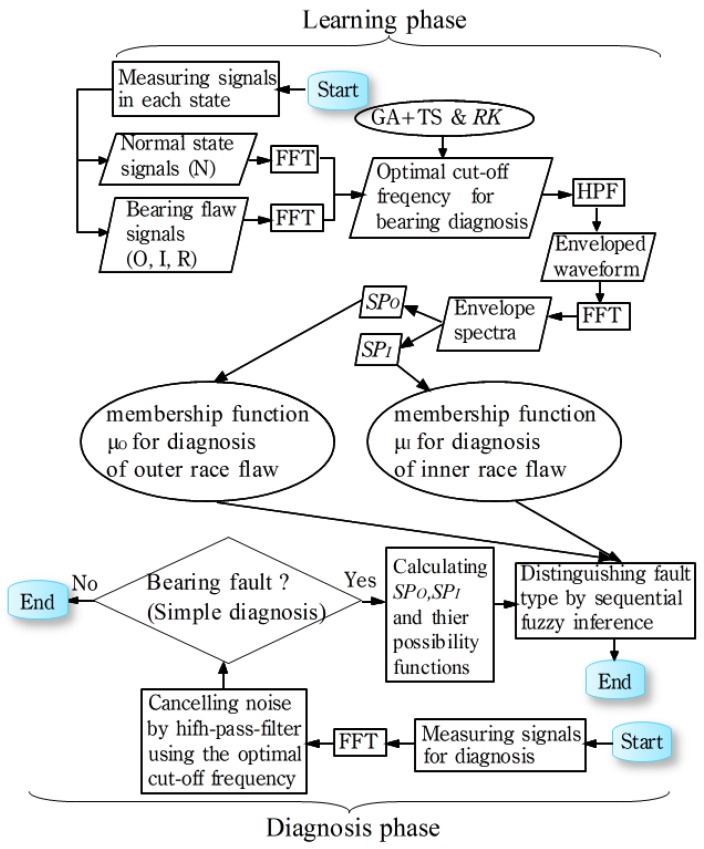
Detail flowchart of the proposed bearing fault isolation.

**Figure 3 sensors-19-03553-f003:**
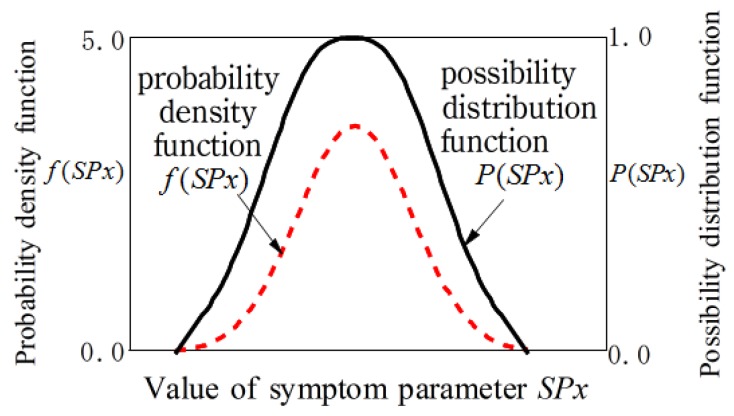
An example of probability possibility function and density function.

**Figure 4 sensors-19-03553-f004:**
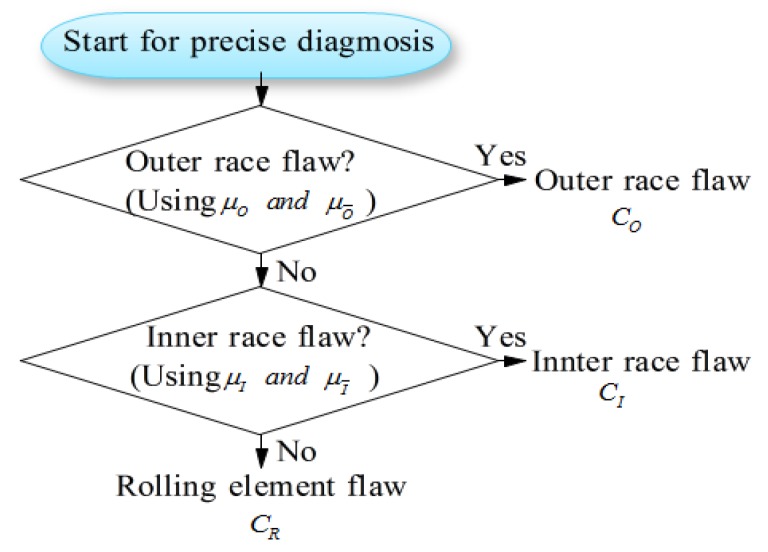
Flowchart of fuzzy inference for bearing diagnosis.

**Figure 5 sensors-19-03553-f005:**
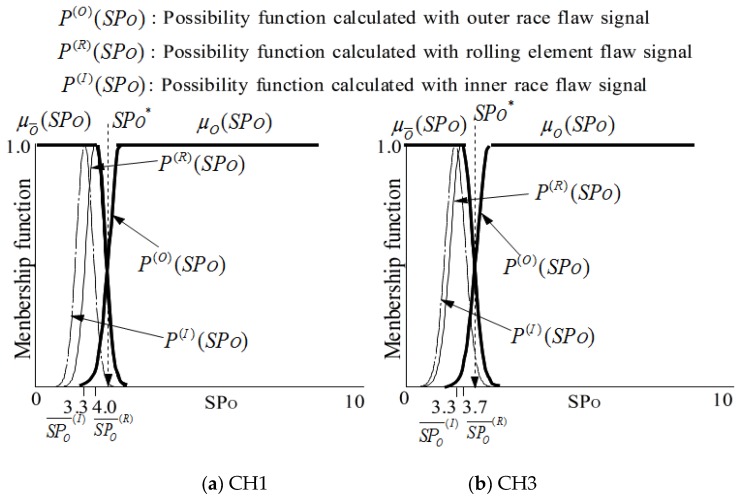
Membership functions $\mu_{O}\left(SP_{O}\right)$ and $\mu_{\bar{O}}\left(SP_{O}\right)$ for recognition of outer race flaw.

**Figure 6 sensors-19-03553-f006:**
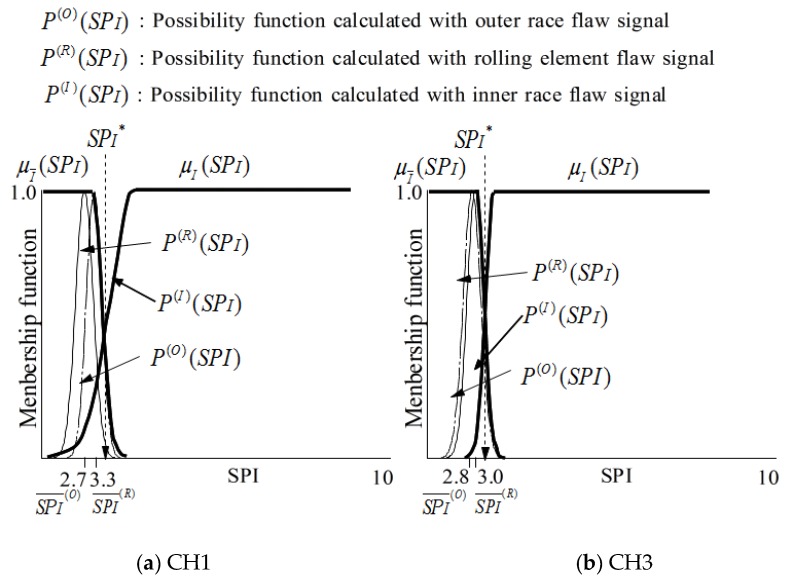
Membership functions $\mu_{I}\left(SPI\right)$ and $\mu_{\bar{I}}\left(SPI\right)$ for recognition of inner race flaw.

**Figure 7 sensors-19-03553-f007:**
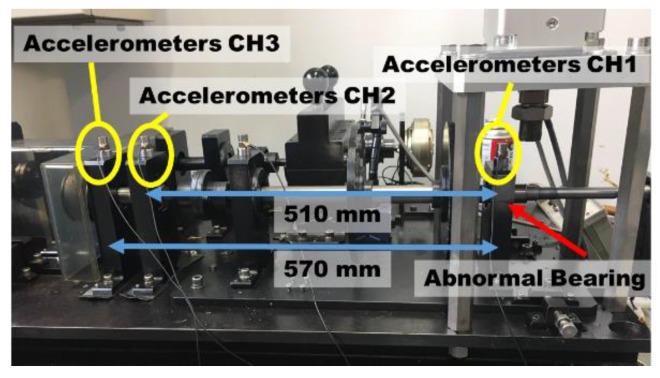
Rotating machine for experimental equipment simulating as a pump facility.

**Figure 8 sensors-19-03553-f008:**
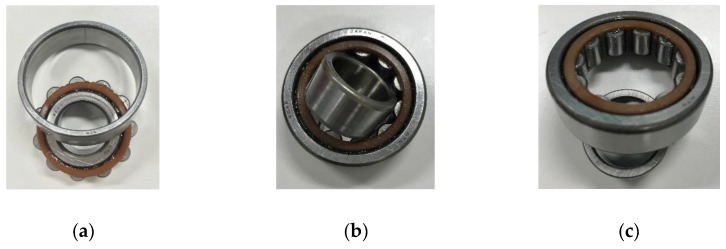
Example of bearing faults. (**a**) Outer race flaw. (**b**) Inner race flaw. (**c**) Rolling element flaw.

**Figure 9 sensors-19-03553-f009:**
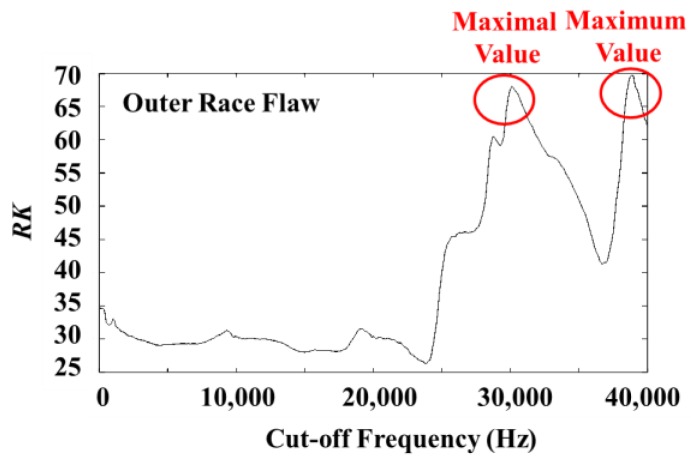
Change curve of cutoff frequency and RK in outer race flow data at CH1.

**Figure 10 sensors-19-03553-f010:**
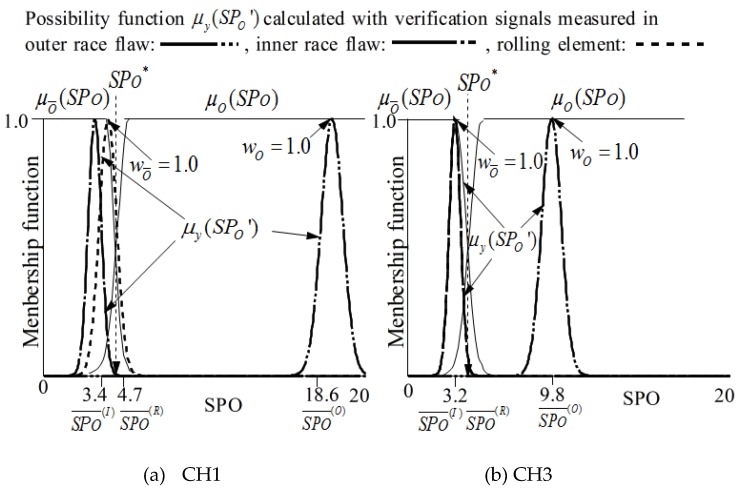
$\mu_{O}\left(SP_{O}\right)$, $\mu_{\bar{O}}\left(SP_{O}\right)$, and $\mu_{y}$ (*SP*_O_*’*) for the judgement of outer race flaw.

**Figure 11 sensors-19-03553-f011:**
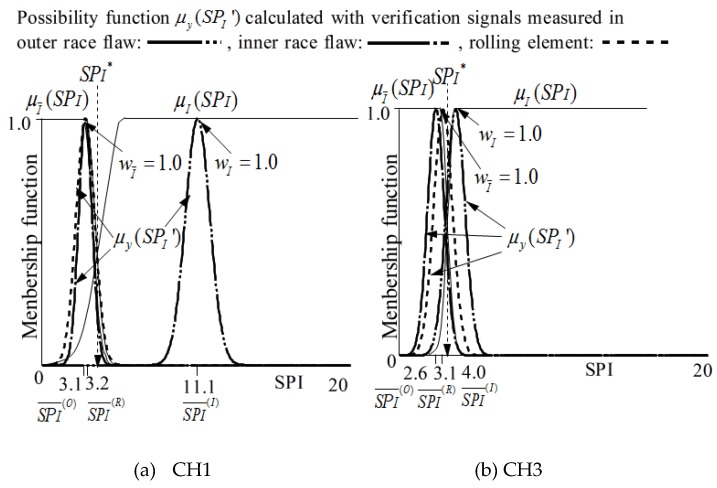
$\mu_{I}\left(SP_{I}\right)$, $\mu_{\bar{I}}\left(SP_{I}\right)$, and $\;\mu_{y}$ (SPO’) for the judgement of inner race flaw.

**Figure 12 sensors-19-03553-f012:**
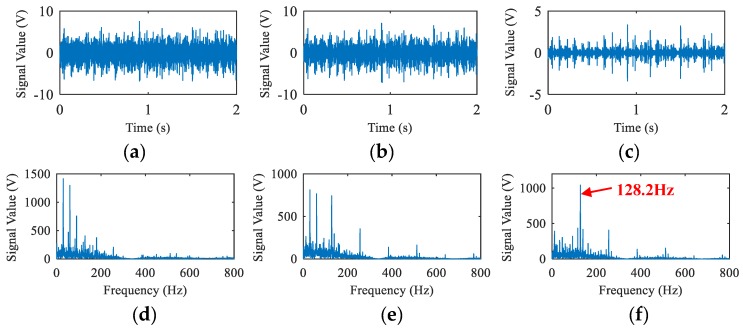
Time waveforms and envelop spectra of outer race flaw signal at 1800rpm (CH3). (**a**) Waveform of the raw data. (**b**) Waveform of the signal after high-pass filter with cutoff-frequency 1KHz. (**c**) Waveform of the signal after high-pass filter with the optimized cutoff frequency. (**d**) Envelop spectrum of the raw data. (**e**) Envelop spectrum of the signal after high-pass filter with cutoff-frequency 1KHz. (**f**) Envelop spectrum of the signal after high-pass filter with the optimized cutoff frequency.

**Figure 13 sensors-19-03553-f013:**
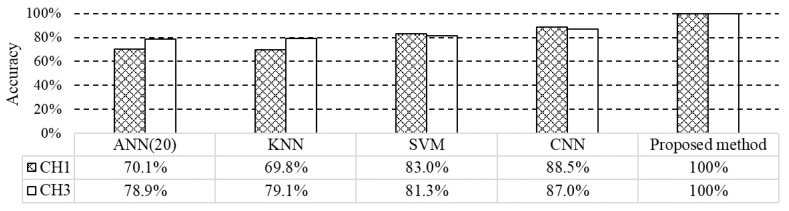
Comparison of recognition accuracies.

**Table 1 sensors-19-03553-t001:** Optimum cutoff frequency and kurtosis ratio (RK) of CH3 (learning phase).

State of the Bearing	Optimum Cutoff Frequency	*RK*
normal	23.4 kHz	1.02
outer race flaw	22.6 kHz	1.35
inner race falw	31.0 kHz	5.08
rolling element flaw	36.1 kHz	1.65

**Table 2 sensors-19-03553-t002:** Optimum cutoff frequencies and RK*s* in outer race flaw data at CH1.

Processing	Optimum Cutoff Frequency	Maximum *RK*
before GA	0 Hz	34.58
afetr GA	30.1 kHz	67.81
after hybrid-GA	38.8 kHz	70.14

**Table 3 sensors-19-03553-t003:** The dedicated symptom parameters for bearing fault isolation of CH3 (learning step).

Fault Type	*SPO*	*SPI*	*SPR*
Outer race flaw 1	7.4342	2.3796	2.9357
Outer race flaw 2	4.8949	1.5213	2.3050
**…**	**…**	**…**	**…**
Outer race flaw 8	6.6856	3.1085	3.7700
Inner race flaw 1	4.0626	3.0314	1.4326
Inner race flaw 2	2.1924	2.9479	2.5816
**…**	**…**	**…**	**…**
Inner race flaw 8	3.9636	4.3204	3.4328
Rolling element flaw 1	4.9868	3.1679	2.5710
Rolling element flaw 2	4.1288	1.7507	2.8610
**…**	**…**	**…**	**…**
Rolling element flaw 8	3.0087	3.0356	2.7371

**Table 4 sensors-19-03553-t004:** $\bar{SPx}$ and $\sigma_{SPx}$ calculated in each state of CH3 for learning.

Signal State	*SPO*		*SPI*		*SPR*	
	${\bar{SPO}}^{\left(x\right)}$	$\sigma_{SPO}{}^{\left(x\right)}$	${\bar{SPI}}^{\left(x\right)}$	$\sigma_{SPI}{}^{\left(x\right)}$	${\bar{SPR}}^{\left(x\right)}$	$\sigma_{SPR}{}^{\left(x\right)}$
Outer race flaw signal	6.6	1.0	2.8	0.6	3.1	0.8
Inner race flaw signal	3.3	0.8	4.3	1.1	2.6	0.7
Rolling element flaw signal	3.7	1.0	3.0	0.5	2.8	0.5

**Table 5 sensors-19-03553-t005:** Optimum cutoff frequency and RK (verification step).

State	*Cut-Off Frequency*	*RK*
Normal	785 Hz	1.09
Outer race flaw	20.5 kHz	1.78
Inner race flaw	33.4 kHz	1.73
Rolling element flaw	32.1 kHz	2.41

**Table 6 sensors-19-03553-t006:** DSP for bearing fault isolation (verification step).

Fault Type	*SPO*	*SPI*	*SPR*
Outer race flaw 1	10.128	1.9491	2.8082
Outer race flaw 2	10.298	1.6400	2.8070
**…**			
Outer race flaw 8	8.2024	2.3197	2.4672
Inner race flaw 1	3.2607	4.1892	3.9748
Inner race flaw 2	2.9917	4.6307	2.6820
**…**			
Inner race flaw 8	3.0646	5.5968	2.5874
Rolling element flaw 1	2.8921	1.9803	3.3640
Rolling element flaw 2	2.7422	4.1339	2.8054
**…**			
Rolling element flaw 8	3.3151	2.7071	2.9271

**Table 7 sensors-19-03553-t007:** $\bar{SPx}$ and $\sigma_{SPx}$ calculated in each state of CH3 for verification.

Signal state	*SPO*		*SPI*		*SPR*	
	${\bar{SPO}}^{\left(O\right)}$	$\sigma_{SPO}{}^{\left(O\right)}$	${\bar{SPI}}^{\left(O\right)}$	$\sigma_{SPI}{}^{\left(O\right)}$	${\bar{SPR}}^{\left(O\right)}$	$\sigma_{SPR}{}^{\left(O\right)}$
Outer race flaw signal	9.8	0.9	2.6	0.7	2.6	0.3
Inner race flaw signal	3.1	0.3	4.0	0.9	2.9	0.5
Rolling element flaw signal	3.2	0.3	3.1	0.9	2.7	0.4

**Table 8 sensors-19-03553-t008:** Diagnosis results based on the Rule 1 of fuzzy inference

State of Verification Signal	$\mathrm{Possibility}\;\mathrm{by}\;\mu_{O}\left(SPO\right)\;$	$\mathrm{Possibility}\;\mathrm{by}\;\mu_{\bar{O}}\left(SPO\right)$	Judge
O	$w_{O}=1.0$	$w_{\bar{O}}=0.0$	O
I	$w_{O}=1.0$	$w_{\bar{O}}=1.0$	$\bar{O}$
R	$w_{O}=1.0$	$w_{\bar{O}}=1.0$	$\bar{O}$

**Table 9 sensors-19-03553-t009:** Diagnosis results based on Rule 2 of fuzzy inference

State of Verification Signal	$\mathrm{Possibility}\;\mathrm{by}\;\mu_{O}\left(SPO\right)\;$	$\mathrm{Possibility}\;\mathrm{by}\;\mu_{\bar{O}}\left(SPO\right)$	Judge
I	$w_{I}=1.0$	$w_{\bar{I}}=0.0$	I
R	$w_{I}=0.0$	$w_{\bar{I}}=1.0$	R

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
