# Peer review of "Automatic Fault Detection and Isolation Method for Roller Bearing Using Hybrid-GA and Sequential Fuzzy Inferenceâ€"

_sensors, 2019, doi:10.3390/s19163553_

Round 1
Reviewer 1 Report
The Authors present a Fault Detection and Isolation method for the diagnosis of bearings. Two distinct methods are proposed for the two phases of the diagnosis. For the fault detection they use the Kurtosis calculated on the test signal compared with the one corresponding to the health component. The value of the ratio depends on the cutoff frequency of a high pass filter. this cutoff is optimized by means of a genetic algorithm combined with tabu search. the authors demonstrate that this approach allows to defining a robust threshold for the fault detection. The isolation of the fault is performed by means of a fuzzy classifier, trained with a suitable number of examples.
Remarks:
>Rows 49-50: it is not clear if the aim of the authors was to define a diagnostic system independent on the tested device. if not, it is difficult to understand the first problem
>Row 69 "Dedicated symptom parameter (DSP)": perhaps there is no need to define an acronym for this, especially because this one is already commonly used in signal processing
>Row 72-74: please re-write the point 4)
> Row 76: "2.1" repeated
> Rows 82-83 "which ... step": please check the sentence
> Row 87 "Condition surveillance": there is a huge literature on diagnosis, with a consolidated terminology. my suggestion is to adopt conventional terms as far as possible. In particular, this action is called fault detection ...
> Row 94: ... so as the "precise diagnosis" corresponds to fault isolation. If it is not, please clarify the difference
> Rows 106-107 " In the condition ... is abnormal": please check the sentence
> Par. 2.2: it is not clear if the optimization of the cutoff is performed any time the diagnosis is applied
> Row 131: (normal, ...): please describe how these examples are collected, how much the indicators are robust with respect to the variability within a class fault, how representative is the training set of the entire possible population of faults
> Rows 305-307: please describe how the comparison has been performed. In particular, if the authors trained the compared systems or they assumed the performance retrieved by the literature. if the case is the second, please indicate how the similarity of the databases has been evaluated
> References: some classical title on FDI could improve the paper
Author Response
Dear Reviewer,
Thank you very much for your decision and kind helps. We have thoroughly revised the manuscript taking into consideration of the useful comments from the Editor and the reviewers. The revised contents are shown by red characters in the response document and revised manuscript.

Reviewer 2 Report
Regarding the paper entitled "Fault Auto-detection Method of Roller Bearing Using Hybrid-GA And Sequential Fuzzy Inference". The paper is quite interesting and presents a good proposal by locating the accelerometers far from the diagnosed bearing. Nevertheless, there are some remarks to be considered in the revised version:
* The state of the art in the introduction is too short. Some papers regarding different methods for fault detection based data-driven and model-based techniques should be considered, for instance:
>https://doi.org/10.1016/j.ifacol.2018.09.604
>https://doi.org/10.1016/j.microrel.2017.03.006
>DOI: 10.1109/TCSII.2019.2920609
*It is not clear for me, what are the conditions for a real implementation (online, real-time), due to the fact the multiple steps of the proposed method, which can be easily verified in the block diagrams.
*I Would like to see the lectures of the accelerometers with and without filtering. Only with the spectrum plots, I cannot verify if the signals are highly noised as stated by the authors. In any case, It is possible to see the vibration signals before and after the fault appearance.
Author Response

(The authors gave the same response as above.)

Round 2
Reviewer 2 Report
Review reference, some are incomplete (e.g. ref. 11).
I don't have more comments regarding the main contribution.